# Phenanthroline-carbolong interface suppress chemical interactions with active layer enabling long-time stable organic solar cells

Xue Lai,[1,2,5] Shiyan Chen, [2,5] Xiaoyu Gu,[3] Hanjian Lai,[2] Yunpeng Wang,[2] Yulin Zhu,[2] Hui Wang,[2] Jianfei Qu,[2] Aung Ko Ko Kyaw [3], Haiping Xia [2,4] ✉ & Feng He [2,4] ✉

To restrain the chemical reaction at cathode interface of organic solar cells, two cathode interfacial materials are synthesized by connecting phenanthroline with carbolong unit. Consequently, the D18:L8-BO based organic solar cell with double-phenanthroline-carbolong achieve the highest efficiency of 18.2%. Double-phenanthroline-carbolong with larger steric hindrance and stronger electron-withdrawing property confirms to suppress the interfacial reaction with norfullerene acceptor, resulting the most stable device. Double-phenanthroline-carbolong based device can sustain 80% of its initial efficiency for 2170 h in dark $N_2$ atmosphere, 96 h under 85 °C and keep 68% initial efficiency after been illuminated for 2200 h, which are significantly better than bathocuproin based devices. Moreover, superb interfacial stability of double-phenanthroline-carbolong cathode interface enables thermal posttreatment of organic sub-cell in perovskite/organic tandem solar cells and obtained a remarkable efficiency of 21.7% with excellent thermal stability, which indicates the potentially wide application of phenanthroline-carbolong materials for stable and efficient solar device fabrications.

Organic solar cells (OSCs) attract increasing attention for merits like printable fabrication processing, lightweight, flexible and wearable nature[1,2]. Due to broad absorption to red and infrared light and efficient electron transport, the nonfullerene acceptors (NFAs) based OSCs boosted efficiency from 6.8% in 2015[3] to currently, over 19%[4,5]. However, the long-term stability under working conditions has always been a major obstacle for their further commercialization[6].

The sequential improvement of OSCs has benefitted from the innovation of highly efficient polymer donors and low bandgap NFAs which can utilize more photons and maintaining a small voltage loss[3,7]. Different from the fullerene and its derivatives, NFAs normally have an acceptor-donor-acceptor (A-D-A) structure with a C = C linker between the donor and acceptor moieties to maintain the conjugated structure. However, the C = C bonds in the NFAs are vulnerable sites that are prone to react with low work function interlayers[8,9] and basic materials[10]. Generally, many classical and highly efficient cathode interfacial materials (CIMs) such as polyethylenimine (PEI), polyethylenimine ethoxylated (PEIE) and poly[(9,9-bis(30-(N,N-diethylamino)propyl)−2,7-fluorene)-alt-2,7-(9,9-dioctyl-fluorene)] (PFN) contain amine groups in their structures. The lone-pair electron in amines can function as a nucleophile with NFAs through an addition reaction and eventually change the large-conjugated structure of the NFAs, resulting in failure of the photovoltaic characteristic[11,12] and stability of the corresponding device during the long-term operation. Considering

[1]School of Chemistry and Chemical Engineering, Harbin Institute of Technology, Harbin 150001, China. [2]Shenzhen Grubbs Institute and Department of Chemistry, Southern University of Science and Technology Shenzhen, Shenzhen 518055, China. [3]Guangdong University Key Laboratory for Advanced Quantum Dot Displays and Lighting, and Department of Electrical & Electronic Engineering, Southern University of Science and Technology, Shenzhen 518055, China. [4]Guangdong Provincial Key Laboratory of Catalysis, Southern University of Science and Technology Shenzhen, Shenzhen 518055, China. [5]These authors contributed equally: Xue Lai, Shiyan Chen. ✉e-mail: xiahp@sustech.edu.cn; hef@sustech.edu.cn

the potential risk of efficiency loss arise from chemical structure altering of NFAs, the development of high-performance CIMs with improved chemical and structural stability is a feasible strategy to prolong device stability. Many efforts have been carried out to overcome this interfacial instability by developing more stable CIMs to minimize the reactivity of the amine groups in CIMs. Xiong et al.[9]. proposed a method to protonate the amine group in PEIE to discourage the interfacial reaction and achieved a flexible NFA OSC with a power conversion efficiency (PCE) of 12.5%. Qin et al.[13] found chelation of PEI with $Zn^{2+}$ (PEI-Zn) can be realized by addition of zinc acetate dihydrate into the PEI solution. This allows strong chelation between $Zn^{2+}$ and amino groups, thus inhibiting the reaction between PEI and the IT-4F. In this way, chelation of metallic elements was shown to be another way to design and synthesize high-performance CIMs with good device efficiency and stability.

Among all the efficient CIMs, the alcohol-soluble small molecular are the most popular due to the convenient synthesis route synthesis and easy functional group modification. Bathocuproine (BCP) is the first reported organic CIM to facilitate electron transport in OSCs[14]. Subsequently, multiple studies of BCP and its derivatives have been reported to further improving the device efficiency and stability, because a wide-bandgap CIM with a deep HOMO energy level is favorable for OSCs due to the sufficient electron transport. BCP is mainly hampered by its structural instability and chemical reaction with NFAs as has been reported previously[15]. 1,10-Phenanthroline (Phen), with its rigid and planar chemical structure is a versatile chelating agent which can stabilize low-valent metals[16]. Consequently, the design and development of morphologically stable CIMs based on Phen can simultaneously improve electronic transport and simplify their synthesis.

Interfacial engineering has been proven as an efficient strategy to improve the efficiency of OSCs. Moreover, the highly efficient interfacial materials with suppressed chemical reactivity with active layer and inherent stability not only benefits the lifetime of single junction solar cells, but can also work well as an interconnecting layer (ICL) in OSC-based tandem solar cells (TSC) that enable a thermal post-treatment process for top OSC photon-absorption layer, which is a process that researchers commonly tried to avoid in previous reports[17–19]. Herein, based on previous study of carbolong materials on solar cell field[20–22], we demonstrate two highly efficient, alcohol-soluble CIMs, a single-phenanthroline-carbolong (SPC) and a double-

phenanthroline-carbolong (DPC), with the same Phen core connecting one or two carbolong complexes, respectively. Both SPC and DPC exhibit decreased HOMO and LUMO energy levels and enhanced electrical conductivity than BCP, which are beneficial for efficiency enhancement of the corresponding devices. A work function shift of the Ag cathode closer to the LUMO of the acceptor was also observed. As a result, the DPC and SPC CIL based devices achieve excellent efficiencies of 18.2% and 17.8%, respectively, which are higher than the control BCP-based device (17.4%). More importantly, due to the strong electron-withdrawing property and the large steric-hindrance of DPC, the chemical reaction between CIL and acceptor (L8-BO in this work) is restrained. Meanwhile, the DPC with good structural stability can prohibit the diffusion of the photoactive material to the electrode for its robust structure. Therefore, the DPC OSC shows an enormous enhancement of device stability that retained 80% of initial efficiency ($T_{80}$ lifetime) for 96 h after being heated at 85 °C while the BCP-based device fast decreased to almost 0% within only 1 h. Besides, DPC-based device maintained a $T_{80}$ lifetime about 2170 h when kept in dark while the BCP contender device quickly dropped to 5% of its initial PCE within 195 h. Moreover, the DPC-based OSC showed significantly improved illumination stability that can maintain 68% after being illuminated for 2200 h. More than that, the DPC with improved photovoltaic performance and stability, can work well as an ICL for perovskite/OSC tandem solar cell and enable the top OSC cell undergo a thermal annealing post-treatment ultimately achieved a high efficiency of 21.7%. This work provides a universal methodology of CIMs design and synthesis utilizing the idea of chemical inserting between CIM and NFAs thus benefit for application in photovoltaic device.

## Results

### Materials synthesis and characterization

As outlined in Fig. 1, the compounds of SPC and DPC were synthesized according to an addition reaction of 3-ethynyl-1,10-phenanthroline (Phen-S1) or 3,8-bis(ethynyl)−1,10-phenanth-roline (Phen-D1) with a carbolong complex (S1), which yielded a productivity more than 90% in a one-step synthesis route. It is the electrophilic addition of carbyne by the alkyne under the synergistic effect of $H^{+22}$. The structures of SPC and DPC were characterized by nuclear magnetic resonance (NMR) spectroscopy and electrospray ionization mass spectrometry (ESI-MS) (Supplementary Figs. 1–8). We successfully obtained the single crystal of SPC (Supplementary Fig. 9), and the detailed crystal data is provided

**Fig. 1 | Synthesis method.** The one step synthesis routes with yields over than 90% of (**a**) SPC and (**b**) DPC.

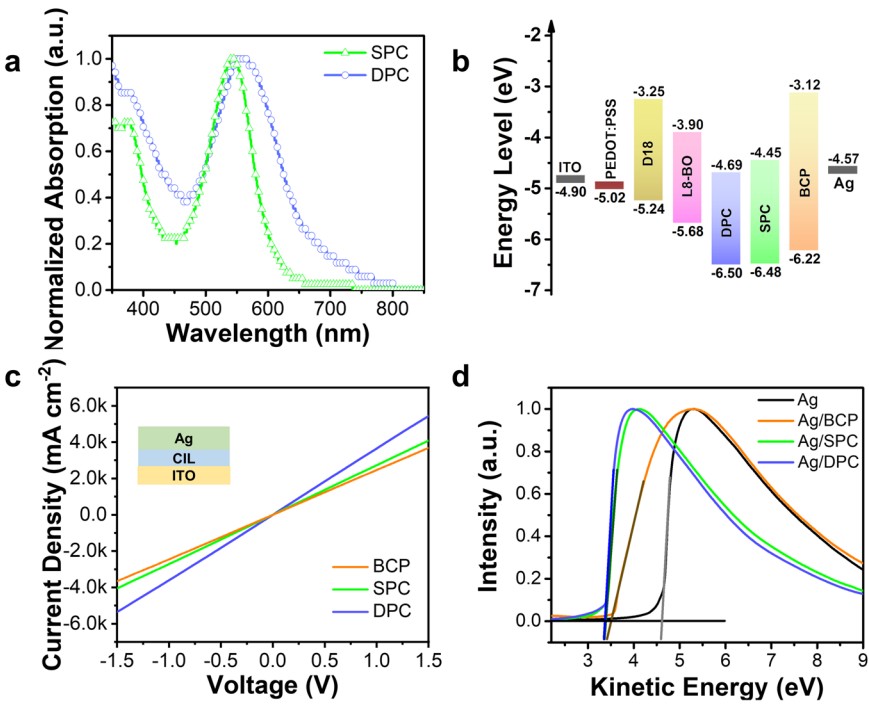

**Fig. 2 | Materials characterization. a** The normalized absorption spectra of SPC and DPC film. **b** The energy level of each functional layer in OSC. **c** The conductivity of various CILs measured by *I-V* curves with a sandwich structure of ITO/BCP/Ag, ITO/ SPC/Ag and ITO/DPC/Ag. Insert shows the structure used in this test. **d** UPS spectra of Ag with or witout being coated with BCP, SPC and DPC CIL.

in Supplementary Table 1. Similar to the structure of BCP, both SPC and DPC have a Phen core but connecting one or two carbolong complex units (Supplementary Fig. 10). Due to the strong electron-withdrawing properties and large steric hindrance of the carbolong unit, the photovoltaic properties and structural stability of both SPC and DPC have been greatly improved when compared with BCP that will be discussed later.

The photovoltaic properties of CIL can directly influence the efficiency of OSCs. Systematic studies were carried out to investigate the effect of the carbolong substitution on the photovoltaic properties of CIMs and compared them with BCP. Fig. 2a shows the normalized ultraviolet visible (UV-vis) absorption spectra of SPC and DPC. The absorption peak of SPC is located at 543 nm and the peak of DPC red shifted to 558 nm because of the added carbolong substitution. The optical bandgap ($E_g$) of these three materials can be obtained from the Tauc plot according to the following equation[23]: $\alpha h\nu = A(h\nu - E_g)^{1/2}$, where A is a content, $\alpha$ is the absorption coefficient and $h\nu$ is the photon energy. Linear fits of $(\alpha h\nu)^2$ curves of BCP, SPC and DPC (Supplementary Fig. 11a) given band gaps of 3.10, 2.03 and 1.81 eV, respectively. The detailed parameters are summarized in Table 1. The highest occupied molecular orbital (HOMO) energy levels of BCP, SPC and DPC are measured by ultraviolet photoelectron spectrometry (UPS) measurement (Supplementary Fig. 11b, c) and were determined to be −6.22, −6.48 and −6.50 eV, respectively. The lowest unoccupied

molecular orbital (LUMO) energy levels of BCP, SPC and DPC obtained by the $E_g$ and HOMO energy level which calculated to be −3.12, −4.45 and −4.69 eV, respectively. The simulated distribution of HOMO and LUMO orbitals was provided in Supplementary Fig. 12. There is strong π-delocalization between the Phen core and carbolong frameworks in SPC and DPC. The energy levels of each functional layer in the device are exhibited in Fig. 2b and Table 1, meanwhile the energy level of other layers are derived from previous reports[24,25]. When compared with BCP, the LUMO energy levels of SPC and DPC decreased by 1.33 and 1.57 eV, respectively, closer to the LUMO energy level of L8-BO acceptor and hence benefit the electron transportation at the active layer/ETL interface. Besides, the HOMO energy levels of SPC and DPC CIL are slightly less than that of BCP which can more efficiently block the injection of holes from the active layer to the cathode thus suppressing the recombination. We summarized the energy levels of highly efficient NFAs, commonly used CIMs as well as metal electrodes that have been reported in recent years (Supplementary Fig. 13 and Supplementary Table 2). Both deep HOMO and LUMO energy levels of SPC and DPC are more suitable with the novel NFAs in current OSCs. For OSCs, the CIL is regarded as an exciton separation and charge transport layer at the cathode. To understand the carbolong substitution on conductivity of SPC and DPC which is directly relate to the device performance according to efficient electron transmission. The self-doping effect of CIMs were investigated by the electron spin resonance (ESR) spectroscopy that can been seen from Supplementary Fig. 14. We firstly investigated the sole component of BCP, SPC and DPC. Very weak signal around g value of 2 from unpaired electron were detected in BCP, SPC and DPC. To further resealing the self-doping effect happed between acceptor and CIM, we also tested the ESR spectra of L8-BO:CIMs. The sample was prepared by dissolving L8-BO and CIM with molar ratio of 1:1 in chloroform followed by drying in a vacuum chamber. From the result, the L8-BO:DPC inhabits the highest resonance peak, suggesting the most efficient electron transport when compared with L8-BO:SPC and L8-BO:BCP. Therefore, we measured the current-voltage (*I-V*) characteristic of sandwiched deice with

**Table 1 | The photophysical properties of BCP, SPC and DPC**

| CILs | Film absorption (nm) | | Conductivity | LUMO | HOMO | WF of Ag with CIL |
|------|------------------------|----------------------|--------------|------|------|-------------------|
| | $\lambda_{max}$ | $\lambda_{edge}$ | mS cm$^{-1}$ | (eV) | (eV) | (eV) |
| BCP | / | 400 | 0.0013 | −3.12 | −6.22 | 3.55 |
| SPC | 543 | 611 | 0.0027 | −4.45 | −6.48 | 3.42 |
| DPC | 558 | 685 | 0.0036 | −4.69 | −6.50 | 3.40 |

structure of ITO/CIL/Ag as shown in Fig. 2c and Table 1. the direct current (DC) conductivity ($\sigma$) can be obtained from the slope of *I-V* plot according to the following equation[26]:

$$I = \sigma A d^{-1} V \tag{1}$$

wherein $A$ is area of device (0.046 cm$^2$), and $d$ is the thickness of film. The $\sigma$ values of BCP, SPC and DPC were 0.0013, 0.0027 and 0.0036 mS cm$^{-1}$, respectively. The DPC exhibits the largest $\sigma$ value indicating the highest vertical conductivity which is good for fast transporting the electron as soon as photogenerated carriers are separated.

The CILs are located between the photon absorber layer and metal electrode that have significant implications on the open-circuit voltage ($V_{OC}$) and fill factor (FF) of device, which can maximize the PCE by tuning the electrode work function. The work function of Ag surface covered by CIL free, BCP, SPC or DPC were obtained by the UPS measurement as shown in Fig. 2d and Table 1. The work function of bare Ag is calculated to be 4.57 eV, which is very close to the result in previous reports[27]. After being covered by BCP, SPC or DPC, the work function of Ag decreased to 3.55, 3.42 and 3.40 eV, respectively. The electron mobility of BCP, SPC, DPC we quantified according to the space-charge-limited current (SCLC) method by measuring electron-only devices with a structure of ITO/ZnO/L8-BO/CIL or CIL free/Ag. As shown in Supplementary Fig. 15, pure L8-BO film gave an electronic mobility of $1.4 \times 10^{-4}$ cm$^2$ V$^{-1}$ s$^{-1}$ because of the unsatisfactory electron transport toward the bare Ag. However, the device associated with DPC CIL achieved a fast electron transport rate of $6.9 \times 10^{-4}$ cm$^2$ V$^{-1}$ s$^{-1}$ which is higher than the values for devices based on SPC ($4.8 \times 10^{-4}$ cm$^2$ V$^{-1}$ s$^{-1}$) or BCP ($4.3 \times 10^{-4}$ cm$^2$ V$^{-1}$ s$^{-1}$). When compared with BCP and SPC CIMs, the better capacity of energy level modification, the most efficient separation of excitons and the fastest charge transfer indicate the DPC a most promising CIM to achieve high device efficiency.

## Device performance and stability study

In order to compare the device performance based on different CILs, several binary OSC with a conventional structure of ITO/PEDOT:PSS/D18:L8-BO/CIL/Ag were fabricated as shown in Fig. 3a. The wide-bandgap polymer D18 was selected as the donor and small molecular L8-BO as the NFA is this work. The current density-voltage (*J-V*) curves of the best device based on CIL free, BCP, SPC and DPC are shown in Fig. 3b, and the detailed photovoltaic parameters are summarized in Table 2. From these results, the device without any CIL exhibits the lowest PCE of 13.3% with a FF of 70.5% and a $V_{OC}$ of 0.780 V implying a large transmission resistance and energy barrier inside the device. From the distribution diagram of each photovoltaic parameter (Supplementary Fig. 16), the $V_{OC}$ and FF of devices increased greatly after the device was inserted a CIL. The optimized device by using DPC as CIL (Supplementary Table 3) obtained the highest PCE of 18.2% with an excellent FF of 80.1%, a $J_{SC}$ of 25.09 mA cm$^{-2}$ and a $V_{OC}$ of 0.905 V, which is higher than that BCP-based device (17.4%) and the single carbolong substituted SPC-based device (17.8%). Besides, we also compared the efficiency of DPC-based device with commonly used CIMs. As shown in Supplementary Table 3, the efficiency of DPC-based device is also higher than devices with PDINN (18.0%), PNDIT-F3N (17.6%) and PFN-Br (16.7%), indicating the better compatibility of DPC in organic solar cells. The external quantum efficiency (EQE) spectra of the best devices based on CIL free, BCP, SPC, and the DPC CIL are presented in Fig. 3c. The $J_{SC}$ values integrated from EQE spectra of corresponding devices are 23.61, 24.23, 24.42, and 24.64 mA cm$^{-2}$, respectively. All the devices are mismatched by only 3% with the current density value from the *J-V* curve.

For a conventional OSC, the CIL is located between the active layer and the metal electrode, thus the chemical and structural stability of the interfacial material can deeply influence the lifetime of the device. The reason the costly and highly efficient BCP rarely used in nonfullerene based OSCs is because the lone electron pair on nitrogen can react as a nucleophile with the C=C to modify the chemical structure of the A-D-A type NFAs[17]. Figure 3d shows the electrostatic potential (ESP) distribution of BCP, SPC and DPC. The positive ESP cover the most of these three CIMs while the electron-rich region is mainly located around the N atom of Phen. The much small ESP value around the N atoms of DPC indicates that the DPC has the weakest nucleophilic ability and thus greatly reduces the chemical reactivity between the acceptor at the interface. Therefore, to track the reaction degree of BCP or DPC with L8-BO in solution over heating time, we measured the $^1$H NMR spectra of prue L8-BO, BCP, DPC and the mix solution of L8-BO:CIM at a molar ratio of 1 :1 in 1,1,2,2-tetra-chloroethane-d$_2$ (C$_2$Cl$_4$D$_2$) which kept heating under 80 °C for 0, 2, 24, 48 and 96 h, respectively. The result is shown in Fig. 3e and f, the red line is $^1$H NMR spectra of pure L8-BO and the single peak located at 9.18 ppm originates from the protons of the C=C. After the L8-BO:BCP mixed solution was heated for 2 h, the characteristic H signal in the C=C peak split to two peaks indicating that the chemical structure of L8-BO began to destroy during the heating process (Fig. 3e). To investigate the chemical reaction product of BCP and L8-BO, we heated the L8-BO:BCP mixed solution under 80 °C for 96 h then seperated the final reaction product according to a chemical sedimentation method. The structure of the reaction product was characterized by $^1$H NMR and mass spectra as shown in Supplementary Fig. 17 and Fig. 18, indicating it was highly possible that the nitrogen atoms on the BCP react rapidly with the C=C double bond of L8-BO and then destroy its structure. However, such chemical reaction can be greatly suppressed by the DPC when the Phen core connects with two carbolong units in the structure. The same experiment was conducted for DPC and L8-BO as shown in Fig. 3f. Due to the electron-withdrawing properties and large steric hindrance of the carbolong substituents, DPC shows suppressive activity of the reaction with L8-BO, which is confirmed by the fact that the characteristic peak at 9.18 ppm of L8-BO was unaffected by heating at 80 °C for 96 h in a mixture solution.

The outstading chemical stability of CIM benifites to the long-term stability of the corresponding device. We further tracked the degradation trend of OSCs by employing different CIMs aged with or witout a temperature of 85 °C in an N$_2$ atmosphere. The results are shown in Fig. 3g and h. The burn-in degradation trend of OSC we obtained includes two stages that are similar to those in previous studies[6]: the fast degradation at the initial stage is caused by the as-cast optimum morphology in the active layer when it is not at thermodynamic equilibrium and is followed by a slow linear degradation. Due to the excellent chemical stability when associated with L8-BO, the DPC-based device exhibits better thermal stability than SPC and BCP-based devices. As shown in Fig. 3g, after being heated at 85 °C for 96 h in the dark, the DPC-based device still retained 80% of its initial PCE (the PCE dropped to 86% in the first 9 h but dropped only 6% in the last 87 h), while the BCP-based device immediately drops to 1% of its initial efficiency in the first 1 h. Because of a less protected reaction site, the efficiency of the SPC-based device quickly dropped to 58% at the first degradation stage. Subsequently, it maintained a very slow decay rate, similar to the DPC-based device. We also compared the thermally stability of DPC-based device with other widely used CIMs including PDINN, PNDIT-F3N and PFN-Br (Supplementary Fig. 19). The result shows that efficiency of device based on PDINN, PNDIT-F3N and PFN-Br fastly decreased below 80% of their initial PCE when heated under 85 °C for 8 h, and maintained only 62.3%, 19.9% and 17.3%, respectively, after being heated for 96 h. Supplementary Fig. 20 indicates that the better thermal stability of DPC and SPC-based devices, especially in the second degradation stage is due to the slow decay rate of $V_{OC}$ and FF. As a result, the best DPC-based device achieved excellent storage stability which maintained 80% of its initial PCE in the dark about 2170 h, which is about 53 times longer than the BCP-based device that rapidly dropped to 80% within 41 h (Fig. 3h). Compared with BCP CIL,

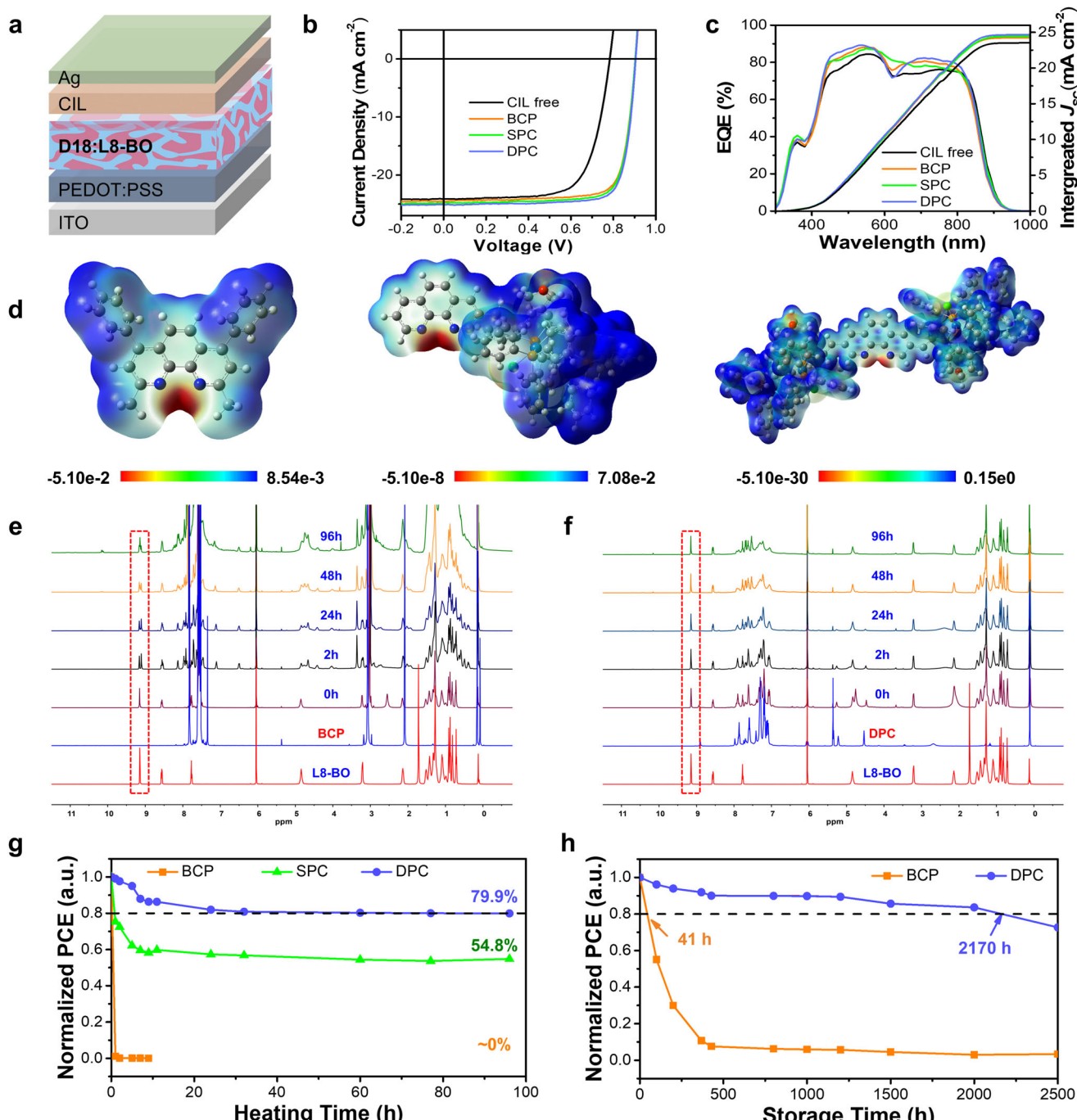

**Fig. 3 | Device performance, cathode interfacial reaction and device thermal study (with or without an 85 °C ageing temperature in N$_2$ atmosphere). a** Device structure of single junction OSC. **b** The *J-V* characteristic curve of the optimized devices based on D18:L8-BO with various CIL, under illumination of AM1.5 G, 100 mW cm$^{-2}$. **c** EQE spectrum of corresponding device. **d** ESP distribution from left to right are BCP, SPC and DPC, respectively. **e** The $^1$H NMR diagram of pure L8-BO, BCP solution, and of L8-BO:BCP mixed solution in C$_2$Cl$_4$D$_2$ after being heated at 80 °C for 0, 2, 24, 48, 96 h. **f** The $^1$H NMR diagram of pure L8-BO, DPC solution, and of L8-BO:DPC mixed solution in C$_2$Cl$_4$D$_2$ after being heated at 80 °C for 0, 2, 24, 48, 96 h. **g** The thermal stability of device based on BCP, SPC and DPC CIL that tested at a 85 °C hotplate for 96 h in N$_2$ atmosphere. **h** The storage lifetime of devices based on BCP and DPC CIL that kept in dark N$_2$ atmosphere for 2500 h.

the slow degradation rate of DPC and SPC, especially at the second stage, implies that the carbolong modification is an efficient strategy with which the CIM can diminish the chemical reaction with NFAs thus increasing the long-term stability of the device.

As OSCs operate under illumination, the photon-degradation is an inevitable barrier to a genuinely stable device. In this work, a white light-emitting diode (LED) with a light intensity of 1 sun was used to investigate the illumination stability of devices based on BCP, SPC and DPC CIL as shown in Fig. 4a. As discussed above, the efficiency aging

tendency of device shows a fast degradation stage followed by a slow one. Because the morphology issue of the active layer mainly happens at the first stage, the efficiency of DPC and SPC-based device dropped to 79.83% and 49.0% of their initial PCE within 301 h, respectively, while that of BCP-based device quickly dropped to less than 5% of its original PCE in 195 h. At the second degradation stage, the decrease of DPC and SPC based devices in PCEs are only about 11.8% and 12.4%, respectively.

To deeply insight into the completely different effect of BCP and DPC on the illumination stability of device, we carried out X-ray

**Table 2 | Photovoltaic parameters of the champion cells with different CILs under AM 1.5 G illumination**

| ETL | $J_{sc}$ (mA cm$^{-2}$) | $V_{oc}$ (V) | FF (%) | PCE$^b$ (%) | $J_{cal,EQE}$$^a$ |
|---|---|---|---|---|---|
| w/o | 24.12 (24.17 ± 0.36) | 0.780 (0.786 ± 0.005) | 70.5 (68.5 ± 1.3) | 13.3 (12.4 ± 0.7) | 23.61 |
| BCP | 24.51 (24.19 ± 0.44) | 0.904 (0.903 ± 0.014) | 78.5 (78.5 ± 0.7) | 17.4 (17.0 ± 0.4) | 24.23 |
| SPC | 24.85 (24.26 ± 0.45) | 0.904 (0.904 ± 0.008) | 79.3 (78.4 ± 0.5) | 17.8 (17.4 ± 0.2) | 24.42 |
| DPC | 25.09 (24.90 ± 0.0.42) | 0.905 (0.904 ± 0.004) | 80.1 (79.4 ± 0.4) | 18.2 (17.9 ± 0.2) | 24.64 |

$^a$The calculated $J_{sc}$ values from EQE curves.
$^b$Average value ± standard deviation was calculated from the statistics at least 10 different devices.

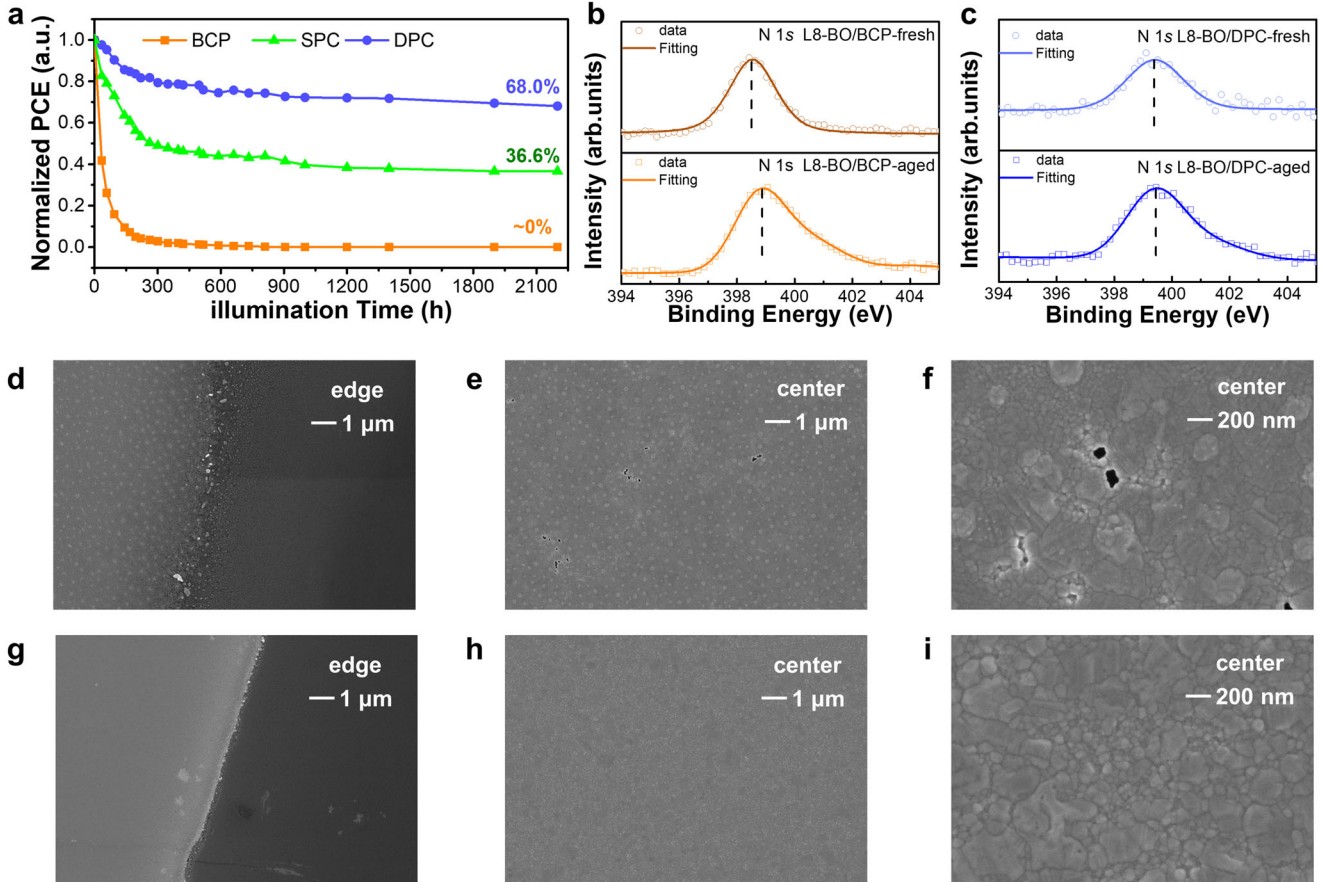

**Fig. 4 | Illumination stability study under a 1 sun LED light for 2200 h in N$_2$ atmosphere. a** Illumination stability of device based on BCP, SPC or DPC CIL that tested for 2200 h. The XPS spectra of N 1$s$ orbital that characterized form the surface of (**b**) L8-BO/BCP film and (**c**) L8-BO/DPC film with or without being illuminated for 2000 h. The edge **d** and center **e-f** SEM images of the Ag electrode upon BCP film after being illuminated for 2000 h. The edge (**g**) and (**h**)−(**i**) center SEM images of the Ag electrode upon DPC-based device after being illuminated for 2000 h.

photoelectron spectroscopy (XPS) measurements to trace the signal of N 1$s$ orbital. In this measurement, we firstly tested the N 1$s$ signal of pure BCP or DPC CIL film without any photoaging as shown in Supplementary Fig. 21. The characteristic N 1$s$ peak of fresh BCP is located at 398.5 eV while DPC shifted to 399.5 eV due to the two carbolong units with strong electron withdrawing property on the structure[28]. For the fresh BCP or DPC film that spin-coated upon L8-BO film as shown in Fig. 4b and c, the XPS peak of N 1$s$ are situated at 398.5 and 399.5 eV, respectively, which are same locations with the pure BCP and DPC film. After been illuminated for 2000 h, the N 1$s$ peak of DPC that covering upon L8-BO layer also show a stronger peak at 399.5 eV which is same location as pure DPC film. However, the peak of L8-BO/BCP film moved to 398.9 eV indicating that the chemical structural of BCP changed during the photon-oxidation process. The negligible influence of L8-BO on DPC CIL under long-term illumination can contribute to the slow PCE decay rate of DPC-based devices. Besides the better chemical

stability of DPC which pacifies the interfacial reaction as discussed above, the DPC with its good structural stability benefits for the blocking of the diffusion of photoactive layer to metal electrode which is another efficiency damage issue occurring in OSCs[29]. To further study the effect of long-time photo-degradation on the Ag electrode, we measured the surface morphology of Ag after the devices had been aged under light for 2000 h. After being photoaged for 2000 h, the edge SEM images of Ag coated upon BCP CIL reveal a large-scale breakdown (Fig. 4d, e) and multiple agglomeration points are shown in Fig. 4f, which is possibly related to the aggregation of active layers or metal diffusion[30]. A large number of pinholes that found on the surface SEM images of BCP/Ag can seriously damage the efficiency of device and lead to an abnormal $J$-$V$ curve[31]. Compared with BCP/Ag, the surface of Ag covered by DPC shows better morphology with less degradation at the edges (Fig. 4g, h) and no agglomeration point or pinholes at the surface as shown in Fig. 4i. Therefore, the negligible

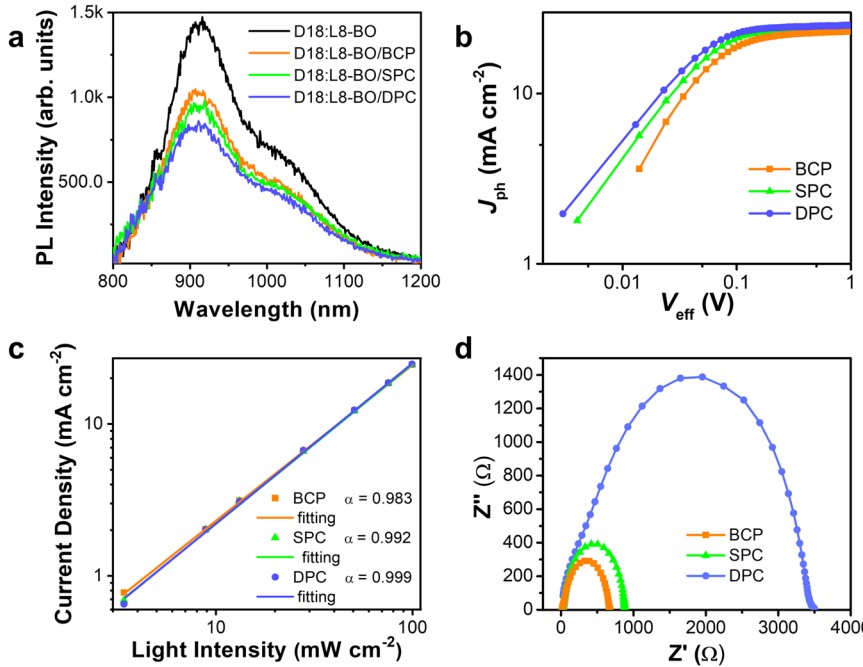

**Fig. 5 | Exciton separation and charge transportation study. a** The PL measurement of D18:L8-BO blends film with or without being covered by BCP, SPC and DPC CIL, respectively. **b** $J_{ph}$-$V_{eff}$ of devices with different CILs. **c** $J_{SC}$ values of the photovoltaic device *vs* light intensity on a double-logarithmic scale. **d** The EIS spectra of device based on different CILs.

reactivity and good structural stability of DPC contributes to the long-time illumination stability of the devices.

## Exciton separation and charge transportation

From the statistics of the device photovoltaic parameters based on CIL free, BCP, SPC, and DPC (Supplementary Fig. 16), the efficiency of DPC-based devices mainly comes from the increase of $J_{SC}$ and FF which account for the efficient collection and extraction of negative carriers[32]. We further confirmed it with steady-state photoluminescence (PL) measurements. The emission peak at 910 nm originates from L8-BO (Supplementary Fig. 22). As shown in Fig. 5a, D18:L8-BO blends covered by DPC CIL exhibit the largest PL quenching efficiency when compared with BCP and SPC, indicating the most efficient charge extraction and transportation process at the active layer/DPC interface. The relationship between photocurrent-density ($J_{ph}$) and effective applied voltage ($V_{eff}$) was tested to investigate the high performance of devices with DPC (Fig. 5b). $J_{ph}$ is calculated as $J_{ph} = J_L - J_D$, where $J_L$ and $J_D$ are the current density under illumination and in the dark, respectively. $V_{eff}$ is determined by $V_{bi} - V_{app}$, where $V_{bi}$ is the built-in voltage and refers to the voltage at which $J_{ph} = 0$ and $V_{app}$ is the applied bias voltage. The charge collection efficiency ($\eta_c$), taking into account both the charge transport in the D18:L8-BO active layer and transfer at the interface could be characterized by the ratio of $J_{ph}/J_{ph,sat}$ under different values of $V_{eff}$, where $J_{ph,sat}$ is the saturated photocurrent at high $V_{eff}$, where the internal electric field is so high that all the photogenerated carriers are swept out to the electrode without recombination, and thus limited only by the absorbed photons. $\eta_c$ of DPC-based device finally calculated to be 0.998 which is higher than the values of SPC (0.997) or BCP (0.996) based device, respectively, suggesting the better charge transport and collection in the DPC-based device which leads to higher $J_{SC}$ and FF. In order to further explore the exciton separation in devices with BCP, SPC or DPC CIM, we conducted transient photocurrent (TPC) measurement as shown in Supplementary Fig. 23. The fitted decay time of BCP, SPC and DPC-based devices are 0.55, 0.23 and 0.22 μs, respectively. The most reduced extraction time confirmed the superior charge extraction ability of DPC-based

device. Fig. 5c shows the curves of $J_{SC}$ as a function of the incident light intensity (I) with the formula of $J_{SC} \propto I^{\alpha}$, in which the deviation of the ideality factor $\alpha = 1$ indicates the degree of bimolecular recombination[33]. From the result, more bimolecular recombination existed in the BCP and SPC-based device with lower $\alpha$ values of 0.983 and 0.992, respectively. For the DPC-based device, an α value of 0.999 reveals the high suppression of bimolecular recombination in the device.

The improved charge dynamics in the DPC-based device can be further confirmed by electrochemical impedance spectroscopy (EIS). The Nyquist plot of EIS measured within the range of 0 to 10 MHz in dark condition under $V_{OC}$ are shown in Fig. 5d and the fitted EIS data according to the equivalent circuit[34] (Supplementary Fig. 24) is listed in Supplementary Table 4. The fitted curves matched well with the original data. $R_1$ accounts for ohmic losses at contacts is in series with charge transport resistance ($R_2$) and recombination resistance ($R_3$)[35]. The DPC OSC exhibits the lowest $R_1$ indicating the smallest contact resistance at the cathode interface. Moreover, the value of $R_3$ significantly increased from 616.1 Ω (BCP device) to 3040.0 Ω (DPC device), indicating the suppressed recombination process in the DPC-based device which is consistence with the improved FF and $J_{SC}$ performance.

## Monolithic perovskite/organic tandem solar cells

Unlike in a regular structure OSC as mentioned above, CIL does not undergo any thermal annealing process. In TSCs with inverted OSC as rear cell, thermal annealing during the deposition of organic active layer will trigger the reaction between the active layer and the fragile ICL, thus leading to significant energy loss in generation and dissociation of excitons[28]. Therefore, inverted OSCs have higher requirements for CILs with thermal and structural stability. Robust DPC material with low chemical activity fits well for inverted OSCs and TSCs, the superb thermal tolerance, structural stability, and feasible orthogonal solvent usage of DPC enables the integration to the ICL in TSCs. We fabricated a monolithic perovskite/organic tandem solar cell with a structure of FTO/ZnO/SnO2/CsPbI2Br/PTAA/MoO3/Ag/ICL

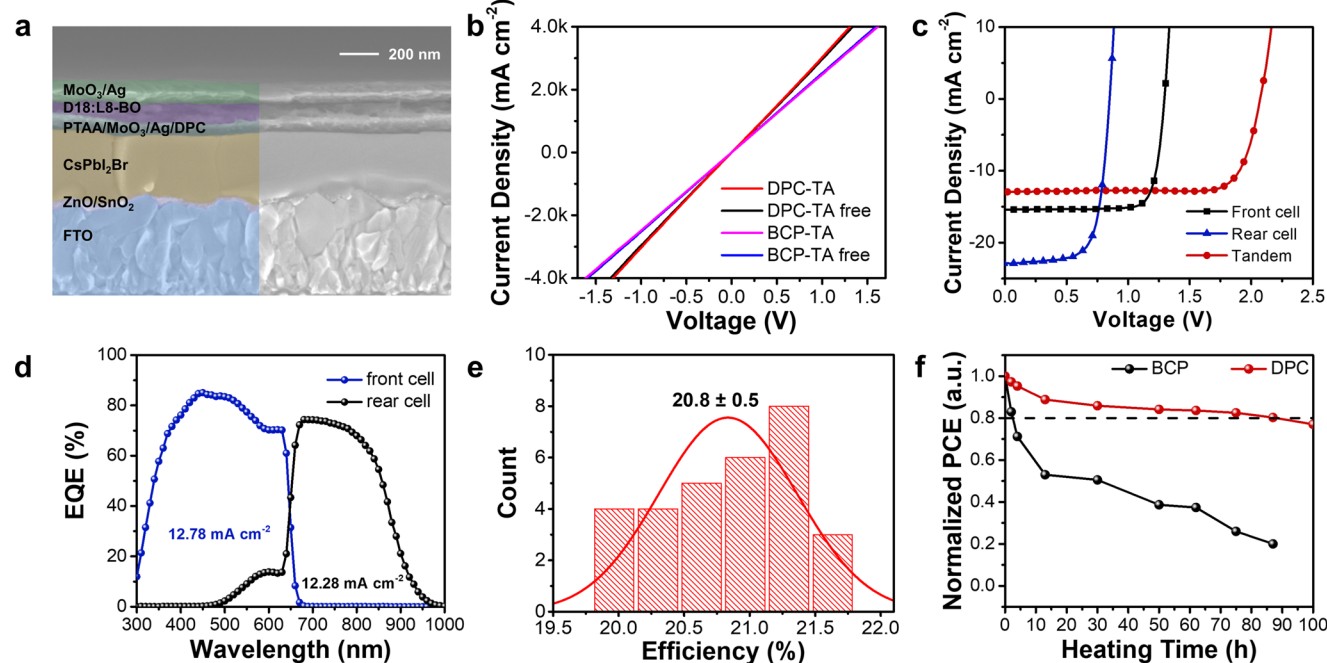

**Fig. 6 | Monolithic Perovskite/Organic Tandem Solar Cells. a** The crossing SEM image of perovskite/OSC tandem solar cell. **b** The conductivity of various CILs measured by *I-V* curves base on structure of MoO₃/Ag/DPC and MoO₃/Ag/BCP with or without a thermal annealing (TA) process at 85 °C. **c** The *J-V* curves of optional individual sub-cells and tandem solar cell, the detailed photovoltaic parameters are summarized in Supplementary Table 5. **d** EQE spectrum of tandem solar cell. **e** The PCE statistic of DPC based tandem solar cells that obtained from 30 cells. **f** The thermal stability of BCP and DPC based TSCs under 85 °C.

/D18:L8-BO/MoO₃/Ag, and the ICL we used here were DPC and BCP. The cross-section SEM image of TSC can be seen in Fig. 6a, where the proper controlled thickness of CsPbI₂Br and organic bulk-heterojunction absorber facilitates the evenly distributed light absorption and current match of sub-cells in TSC device. For perovskite/OSC TSC, the ICL beneath active layer of rear cell should have a high optical transmittance to ensure more transmitted light can be utilized by rear cell. Therefore, we measured the transmittance of BCP and DPC film, with data depicted in Supplementary Fig. 25. Although BCP film has very high transmittance, the DPC film obtained with optimized concentration (1 mg/mL) shows a very high transmittance of more than 97% in the long-wavelength region from 400 nm to 1000 nm, which is good enough for the solar cell utilization. Consequently, the high transmittance of BCP and DPC can ensure adequate light-harvesting for photoactive layer of tandem solar cell. Then, we measured the conductivity of DPC and BCP-based ICL with or without a TA process under 85 °C. As shown in Fig. 6b, DPC-based ICL achieves a higher conductivity than BCP, which can be ascribed to the distinct superior electric property as shown in Fig. 2c. More importantly, the conductivity of DPC ICL increased slightly while the BCP device remained unchanged after thermal annealing under 85 °C for 10 min. The improvement can be ascribed to the increased crystallinity and can serve better bridging the perovskite front cell and organic rear cell and as the recombination center.

The photovoltaic parameters of individual single-junction sub-cells are summarized in Supplementary Table 5. The CsPbI₂Br device, which features an optical bandgap of 1.92 eV, was used to absorb photons with higher energy in tandem devices. Firstly, the BCP and DPC-based devices were prepared with an inverted structure of ITO/ BCP (DPC)/D18:L8-BO/MoO₃/Ag, that were exactly same as rear cell in the tandem device to fully simulate the performance. The result is shown in Fig. 6c and summarized in Supplementary Table 5. BCP device with an inverted structure only obtained a PCE of 3.5% which is much lower than BCP-based OSC with a conventional structure. As a result, the PCE of corresponding BCP-TSC only achieved 12.4%. On the contrary, give the credit to strong electron-withdrawing properties and large steric hindrance of the carbolong units, the DPC device with optimized concentration of 1 mg mL⁻¹ delivered a $V_{OC}$ of 0.85 V, $J_{SC}$ of 22.85 mA cm⁻² with the optimal PCE of 13.7%. More importantly, the resulted DPC-TSC achieved an impressive PCE of 21.7% with a high $V_{OC}$ of 2.07 V, a $J_{SC}$ of 12.95 mA cm⁻² and an astonishing FF of 80.8%. Fig. 6d presents the EQE spectra of DPC-based TSC measured sequentially applying the bias light with 550 nm short pass filter and 800 nm long pass filter. The evenly split spectra of front cell and rear cell generated well matched current owing to the precisely controlled thickness of absorber. Besides, a high average PCE of 20.8% that obtained from 30 cells in Fig. 6e indicates the high repeatability of DPC-based TSC. We further tested the thermal stability of tandem device, and the results are shown in Fig. 6f. Due to the suppressed chemical activity and inherent stability, the DPC-based device maintained a good stability during storage in N₂ glovebox under 85 °C continues heating for 90 h, 80.0% of its initial PCE remained, while the BCP-based device loses its *J-V* characteristic after the same time. The high device performance and enhanced stability of DPC as ICL for tandem solar cells indicating the potential wide application of carbolong in the photovoltaic field.

## Discussion

In this work, we have synthesized two organometallic compounds CIMs demonstrated by combining one (SPC) or two (DPC) carbolong metallic units with 1,10-phenanthroline core as CIMs producing highly efficient OSCs and TSCs. The synthesized CIMs with carbolong substituents exhibit enhanced electron-transport properties and modified work function capability for electrodes, resulting in a high device efficiency of 18.2% based on DPC CIL. More importantly, the introduction of carbolong metallic units in DPC can greatly suppress the reaction with the C = C of nonfullerene acceptor material (L8-BO in this work) and can block the diffusion of photoactive materials at the cathode, which is commonly seen in other amine-containing CIMs. Due to the suppressed chemical activity and inherent stability, DPC maintained excellent device stability when compared to BCP-based device.

The best DPC-based device exhibits a storage $T_{80}$ lifetime of about 2170 h in an $N_2$ atmosphere which is about 53 times longer than the BCP-based device. Meanwhile, the DPC can slow down the device degradation rate and can maintain 68% of its initial PCE after being aged under continuous illumination for 2200 h and retains 80% of its original efficiency after being heated at 85 °C for 96 h. DPC can work well as an interconnecting layer in perovskite/OSC tandem solar cells and achieved a high efficiency of 21.7% which is higher than that of BCP-based device. The DPC-based TSC also can maintain 80% of its initial PCE after being heated under 85 °C for about 90 h. The carbo-long based CIM with efficient photovoltaic performance and a stable structure can ensure the long-time stability and high efficiency of OSCs based on both NFAs and tandem solar cells.

## Methods

### Materials and synthesis

Unless special statement, all materials directly use after purchase including: D18 (Derthon OPV Co LTD), L8-BO (Solarmer materials), BCP (Adamas) and PEDOT:PSS (CLEVIOS™ P VP AI 4083). Solvents were distilled under nitrogen from sodium/benzophenone (hexane, tetra-hydrofuran, diethyl ether) or calcium hydride (dichloromethane, DCM). Other reagents were used as received from Aldrich Chemical Co. Column chromatography was performed on alumina gel (200–300 mesh), silica gel (200–300 mesh) or polystyrene gel (Bio-Beads ™S-X3 Support, 200–400 mesh) in air.

### Characterization of materials

NMR spectroscopic experiments were performed on a Bruker Ascend III 400 ($^1$H, 400.1 MHz, $^{13}$C, 100.6 MHz, $^{31}$P, 162.0 MHz) spectrometer at room temperature. $^1$H and $^{13}$C NMR chemical shifts (δ) are relative to tetramethyl silane (TMS), and $^{31}$P NMR chemical shifts are relative to 85% $H_3PO_4$. The absolute values of the coupling constants are given in Hertz (Hz). Multiplicities are abbreviated as singlet (s), doublet (d), triplet (t), multiple (m), and broad(br). High-resolution mass spectra (HRMS) experiments were recorded on Thermo Fisher Scientific Q-Exactive MS System. The theoretical molecular ion peak was calculated by Compass Isotope Pattern software supplied by Bruker Co. Elemental analyses were performed on a Vario EL III elemental analyzer.

All structures were optimized at the B3LYP[36–38] level of functional theory. Frequency calculations were performed to confirm the characteristics of all the calculated structures as minima. All these structures evaluated were optimized at the B3LYP/6–31 G* level of DFT with an SDD basis[39] set to describe Os atom. All the optimizations were performed with the Gaussian 09 software package[40].

Compound SPC was collected on a Bruker APEX-II CCD diffractometer with Cu Kα radiation (λ = 1.54184 Å). Using Olex2[41]. All the structures were solved using the ShelXT[42] structure solution program using the intrinsic phasing method, and all the structures were refined with the ShelXL[43] refinement package using least-squares minimization. Non-H atoms were refined anisotropically unless otherwise stated. Hydrogen atoms were introduced at their geometric positions and refined as riding atoms unless otherwise stated. All single crystals suitable for X-ray diffraction were grown from a solution of $CH_2Cl_2$ layered with hexane. Further details on the crystal data, data collection, and refinements are provided in Supplementary Table 1. The X-ray crystallographic coordinates for structures reported in this study have been deposited at the Cambridge Crystallographic Data Center (CCDC), under deposition numbers 2220750 (SPC), the Cambridge Crystallographic Data Center via www.ccdc.cam.ac.uk/data_request/cif. The detailed information as shown in Supplementary Data 1. The XPS measurement was tested by ULVAC PHI 5000 Versa Probe III with Al Kα radiation (1486.6 eV). UPS were measurement by d He I (21.22 eV) as the excitation source with an energy resolution of 50 meV.

### Device fabrication and characterization.

Single-junction OSC Fabrication: ITO-coated glasses (15 Ω sq⁻¹) were sequentially cleaned by deionized water, acetone, and isopropanol in an ultrasonic bath. After drying, the ITO substrates were UV treatment for 20 min, PEDOT: PSS film was fabricated by spin-coating at 6000 rpm for 22 s then annealed at 150 °C for 10 min on hotplate. For the active layer, the D18 donor and L8-BO acceptor were mixed in CF solvent at a ratio of 1:1.2 with 0.5% (volume ratio) chloronaphthalene (CN) as an additive, the total concentration is 12 mg/ml. The solution was spin-coated on the ITO/PEDOT: PSS substrate at 3000 rpm for 35 s, then annealing at 100 °C for 10 min. For the CIL, the SPC or DPC was dissolved in methanol with different concentration without any cosolvent. The concentration of BCP is 0.5 mg/ml in methanol. All the CILs were fabricated at 2200 rpm for 30 s on the active layer. Finally, device fabrication was completed by thermal evaporation of a 100 nm Ag.

Perovskite/OSC Tandem Device Fabrication: For the single junction perovskite solar cell, s-ZnO layer was formed by spin coating ZnO sol–gel solution at 3000 rpm for 30 s, then annealing at 170 °C for 1 h. After that, $SnO_2$ was spin coated on ZnO layer at 3000 rpm for 30 s, followed by annealing at 150 °C for 30 min. The $CsPbI_2Br$ precursor was spin coated on $ZnO/SnO_2$ ETL in $N_2$ filled glovebox via a two-step spin-coatin followed a gradient annealing process; the depositing procedure was set to be 1000 rpm for 10 s and 3500 rpm for 25 s. Chlorobenzene was dripped onto the rotating substrate 10 s prior to the end of the program. The samples were immediately transferred to a hotplate and sequentially annealed at 50 °C for 1 min and 240 °C for 1 min. PTAA with concentration of 10 mg mL⁻¹ was spin coated on perovskite at 3000 rpm for 30 s. Finally, $MoO_3$ and Ag electrode were thermally evaporated with controlled thickness of 10 and 100 nm, respectively. For BCP or DPC-based single junction invert OSC, BCP or DPC was deposited on ITO substrate at 2000 rpm for 30 s. The D18:L8-BO (1:1.2) blend film was deposited by using dynamic spin coating at 3000 rpm for 30 s, followed by annealing at 100 °C for 10 min in the case of thermal annealing device. Then, $MoO_3$ and Ag electrode were thermally evaporated with controlled thickness of 10 and 100 nm, respectively. The ICL of $MoO_3$ (≈10 nm)/Ag (≈1 nm)/BCP or DPC layer was formed by evaporating $MoO_3$ and Ag; the BCP or DPC layer was spin coated at 3000 rpm for 30 s. Other procedures can be referred to the fabrication in single junction solar cell.

The current-voltage (J-V) characteristics were measured under AM 1.5 G illumination at 100 mW cm⁻² with a solar simulator (Enlitech, Inc) in the glovebox and calibrated by a silicon cell (Hamamatsu S1133 color, with KG-5 visible filter). For J-V curve measurement, the active area is 0.046 cm². EQE measurements were conducted on devices under short-circuit conditions through a lock-in amplifier under illumination with monochromatic light from a xenon lamp (Enlitech, Inc). Mobility of hole and electron was measured accroding to a field-independent space charge limited current (SCLC) model following the Mott-Gurney law:

$$J = \frac{9}{8}\varepsilon_0\varepsilon_r\mu\frac{V^2}{d^3}$$

wherein J is current, $\mu$ is the mobility, $\varepsilon_0$ is the permittivity of free space, $\varepsilon_r$ is the relative permittivity of the material, d is the thickness and V is the voltage. The electron-only device structure is ITO/ZnO/L8-BO/CIL/Ag.

### Photoluminescence measurement

Steady-state photoluminescence (PL) spectra of films were measured with an FLS920 spectro-fluorimeter (Edinburgh Instruments), whose light source system contains 450 W xenon lamp and a Glan prism. Spectrograph detection used Ge detector with reponse range from 800 to 1700 nm. The data was collected and analyzed by the connected F900 systems software.

## Sability measurement

The devices for thermal-stability test were continuous heating under an 85 °C hotplate in glove box. The storage stability was tested in dark $N_2$ atomospher. Illumination stabiliy was obtained by constantly exposing the device stored in the glove box to an LED light (1 sun). The device area is 0.046 cm$^2$ determined by the designed shadow mask calibrated by optical microscope. The start and end sweep voltage were -0.2 V to 1 V with a step of 0.01 V and a dwell time of 1 ms.

## Reporting summary

Further information on research design is available in the Nature Portfolio Reporting Summary linked to this article.

## Data availability

The authors declare that the source data generated in this study are provided in the Supplementary Information and Source Data file. All source data generated during the current study are available from the corresponding authors upon request. Source data are provided with this paper.

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

## Acknowledgements

This work was financially supported by the National Natural Science Foundation of China (22225504, 21975115, 51903116, 21931002), Shenzhen Fundamental Research program (JCYJ20190809163011543 and JCYJ20200109140812302), Guangdong Provincial Key Laboratory of Catalysis (2020B121201002), Guangdong Innovative and Entrepreneurial Research Team Program (2016ZT06G587), Shenzhen Sci-Tech Fund (KYTDPT20181011104007), and China Postdoctoral Science Foundation (No. 2021M701567). We also appreciate the assistance of SUSTech Core Research Facilities with compound characterization.

## Author contributions

X.L. and S.C. contributed equally to this work. X.L. fabricated the devices and carried out solar cell characterizations, X.G. carried out the SEM measurement and fabricated all inorganic perovskite film for the tandem solar cell. H.L., Y.W., and J.Q. carried out the ESP calculations and mass spectrometry measurement. Y.Z. measured the ESR spectrum. S.C. and H.W. designed and synthesized DPC and SPC. A.K., H.X., and F.H. supervised the project. All authors participated in the discussion of the results and the manuscript writing and approved the final submission.

## Competing interests

The authors declare no competing interests.
