## [Peer Review File · Nature Communications]

Phenanthroline-Carbolong Interface Suppress Chemical Interactions with Active Layer Enabling Long-Time Stable Organic Solar CellsREVIEWER COMMENTS

Reviewer #1 (Remarks to the Author):

Lai et al. reports the synthesis of two new cathode interfacial materials (CIMs) and their applications in nonfullerene organic solar cells. The new CIMs were obtained by connecting Phen core with one or two carbolong complex. The double-phenanthroline-carbolong (DPC) delivered high performance in organic solar cells, and also showed superior chemical stability with the active layer than the traditional BCP under heat and light. The chemical stability is important for the CIM in efficient nonfullerene solar cells. It has also been a challenging task to develop a stable one and in the meantime delivering high photovoltaic performance. The CIM reported in the work will generate high impact for solar cells application and would give a hint for future development of CIM. Thus, I would recommend its publication in Nat Commun. The following comments should be addressed:

1. The new developed CIM of DPC should be compared with a commonly used CIM PDINN. Please comment it is possible for DPC to replace the PDINN.
2. Please comment how about solvent orthogonality of DPC and the organic active layer? Is it possible to fabricate inverted single cells with DPC CIM?
3. How about the conductivity of DPC? Is there a self-doping effect in the DPC?
4. Please also provide illumination data of the tandem devices.
5. DPC has strong absorption in the visible range. How will it influence the device performance?
6. Please comment any specific reason of using the carbolong unit for designing the DCP.

Reviewer #2 (Remarks to the Author):

For the performance of highly efficient and stable organic solar cells cathode interfacial layers play an important role. They transfer the negative charges and block the positive charges at the cathode. In the present manuscript, the authors report on a variety of cathode interface materials that are available for both efficient single junction organic solar cell and tandem solar cell. According to the rational molecular design strategy of carbonyl substituents on the Phen, the photovoltaic performance of resulted DPC and SPC improved significantly, such as higher electrical conductivity, better matching energy levels or suitable work function modification of metal cathode. As a result, the single junction organic solar cell obtained an OPV efficiency of 18.2% which is higher than that of BCP control device.

The authors found that DPC with two carbonyl substitutions can suppress the negative reaction which commonly happens between the cathode interfacial layer and non-fullerene acceptor (NFA) material, leading to enhanced device stability. In particular, the DPC based device maintained 80% of its initial PCE at elevated temperatures (85°C) for 96 h or at room temperature after 2170 h in inert N₂ atmosphere. As the reported DPC material exhibits overall better photovoltaic performance and stability, DPC offers a wide range of potential applications in OPV community.

Just from these novel achievements, I do recommend the manuscript for publication in Nature Communications. However, there are several aspects that have to be changed/clarified/commented prior to final acceptance.

1. There are too many experimental findings listed in the introduction part. Authors should just name the 3 key findings.
2. Materials characterization section: the achieved data should be summarized in one common table (which also allows better comparison of the materials' properties).
3. Fig. 2d: kinetic energy scale in UPS does not give any information if now experimental details are declared. Actually, all relevant data are shown in Fig. S11 (with all required information on determination of the respective values (slopes and crossing with energy axis))
4. HOMO and LUMO orbitals are important for material analysis. The authors may add the simulated frontier molecular orbitals to the supplementary information. The energy level calculation method should be described in the experimental section of the manuscript.
5. Degradation studies (Figure 3). It is not fully clear how relevant the XPS study is. Is it required to show the materials' degradation in the main text. Major conclusion is that BCP shows significantly different composition (according to N1s binding energies), but the conclusions is more speculative.
6. Page 19, line 340: is the variation in the 3 digit really trustworthy ?

7. Results reported on page 22 just report on the device parameters – authors should focus on the most prominent ones required for their discussion/conclusions.

8. Can the authors give any idea why DPC and SPC are advantageous to suppress interfacial reactions when used as ICL ?

9. Authors should use the identical colour code for BCP, SPC and DPC in all figures.

10. Some abbreviations must be explained at their first appearance (NB: no abbreviations in the abstract).

11. The manuscript may require some rewriting/rephrasing and inspection from a native English speaking person. The use of the article “the” sometimes seems arbitrary. The word “meanwhile” is not used in the proper sense. I also found few incomplete sentences and numerous typos. See e.g. Fig. S20 (“Bing Energy”)

12. Line 385: Supplementary “Table 3.” should be corrected to “Table 4”.

Reviewer #3 (Remarks to the Author):

In this work, the authors developed two newly synthesized cathode interfacial materials, DPC and SPC, which consist of a phenanthroline core and a carbonyl unit. The authors investigated their improved photovoltaic performance and utilized them in organic solar cells as well as perovskite-organic tandem solar cells. They used various methods to evaluate and compare the effect of carbonyl CIMs on device efficiency and stability and confirmed that the resulting DPC can minimize the chemical reaction which indeed plays an important role in the long-term stability of corresponding photovoltaic devices. Overall, this work provides lots of valuable information for molecular design. Publication of this work is recommended in Nature Communications after minor revisions.

1) In Fig. 4b, the authors investigate the charge collection efficiency (η_c) of devices based on various cathode interfacial layers. However, the obtained values for DPC, SPC, and BCP are too close to distinctly compare the charge transfer ability at this layer. The authors should provide more evidence on this point.

2) What is the essential reason for the strong electron-withdrawing properties of carbonyl frameworks?

3) Organometallic compounds generally suffer from poor stability. However, why are carbonyl complexes quite stable?

- 4) The addition reaction of alkynes and metal carbynes is interesting. It is better to add some discussion about this reaction.
- 5) Acronyms for nouns should appear where they are first mentioned in the manuscript, including but not limited to BCP, CIM and TPC.
- 6) The energy levels of commonly used materials in OPV field are summarized in supplementary figure 12, references are needed here.
- 7) Page 26 line 461, reference 42 is wrong.
- 8) In Scheme 1, what is the abbreviation of [Os]? please specify.

Reviewer #1 (Remarks to the Author):

Lai et al. reports the synthesis of two new cathode interfacial materials (CIMs) and their applications in nonfullerene organic solar cells. The new CIMs were obtained by connecting Phen core with one or two carbolone complex. The double-phenanthroline-carbolone (DPC) delivered high performance in organic solar cells, and also showed superior chemical stability with the active layer than the traditional BCP under heat and light. The chemical stability is important for the CIM in efficient nonfullerene solar cells. It has also been a challenging task to develop a stable one and in the meantime delivering high photovoltaic performance. The CIM reported in the work will generate high impact for solar cells application and would give a hint for future development of CIM. Thus, I would recommend its publication in Nat Commun. The following comments should be addressed:

We thank reviewer for taking time to review our manuscript and giving insightful and constructive comments for the improvement of the manuscript.

1. The new developed CIM of DPC should be compared with a commonly used CIM PDINN. Please comment it is possible for DPC to replace the PDINN.

Response: We thank the reviewer for the valuable question. To answer this comment, we compared the efficiency of DPC with the commonly used CIMs including PDINN, PNDIT-F3N and PFN-Br. The results are summarized in Table R1 (Supplementary Table 3) in the revised manuscript with same device structure that we applied in our paper. We also added the following discussion in the revised manuscript:

“Besides, we also compared the efficiency of DPC based device with commonly used CIMs. As shown in Supplementary Table 3, the efficiency of DPC based device is higher than devices with PDINN (18.0%), PNDIT-F3N (17.6%) and PFN-Br (16.7%), indicating better compatibility of DPC in organic solar cells.”

Table R1. The photovoltaic parameters of the best performing device employed with different CIMs under AM 1.5 G illumination.

CIM	J_{sc} (mA cm ⁻²)	V_{oc} (V)	FF (%)	PCE (%)	$J_{cal, EQE}$ (mA cm ⁻²)
DPC	25.09	0.905	80.1	18.2	24.64
PDINN	25.48	0.891	79.1	18.0	24.87
PNDIT-F3N	24.92	0.906	78.1	17.6	24.55
PFN-Br	24.13	0.905	76.1	16.7	23.96

The fabricated device with DPC CIM shows improved thermal stability, and the corresponding T_{80} lifetime is significantly extended, as shown in Fig. 2g in the manuscript. The obtained T_{80} lifetime under thermal stress of continuous 85 °C heating is about 96 h, which is much longer than PDINN (9 h), PNDIT-F3N (0.8 h) and PFN-Br (0.3 h) based device (Fig. R1 and Supplementary Fig. 19 in the revised manuscript). The superior photovoltaic and stability performance make DPC a potential CIM material even to replace the widely used CIMs. In the revised manuscript, we also replaced the thermal stability test of PDINO device to PDINN device for being consistent with the efficiency discussion. The corresponding discussion was change to “We also compared the thermally stability of DPC based device with other widely used CIMs including PDINN, PNDIT-F3N and PFN-Br (Supplementary Fig. 19). The result shows that efficiency of devices based on PDINN, PNDIT-F3N and PFN-Br fastly decreased below 80% of their initial PCE when heated under 85 °C for 8 h, and maintained only 62.3%, 19.9% and 17.3%, respectively, after being heated for 96 h.”

Figure R1. Thermal stability (85 °C) of devices based on various CIMs tested in dark N₂ atmosphere.

2. Please comment how about solvent orthogonality of DPC and the organic active layer? Is it possible to fabricate inverted single cells with DPC CIM?

Response: To clarify the solubility of DPC in various solvents, we dissolved BCP, SPC and DPC separately in chlorobenzene (CB) and chloroform (CF). As is shown in the Fig. R2, distinct difference in solubility can be seen, BCP can be fully dissolved in chloroform while remain undissolved particles in methanol (MeOH) for 10 min without any cosolvent. Both SPC and DPC showed very limited solubility in CB, but got improved in CF solvent (although some solid particles are still present after 10 min of dissolution). Both SPC and DPC had excellent solubility in MeOH.

Figure R2. The solubility test of BCP, SPC and DPC in CB, CF and MeOH, respectively.

As suggested by the reviewer, fabrication of inverted single junction cells with DPC CIL is possible if proper deposition process is followed. We have fabricated devices with a structure of ITO/DPC/D18:L8-BO/MoO₃/Ag, with best performing device presented in Fig. 5c labeled as rear cell and summarized data in Supplementary Table 5. The solvent for active layer deposition is chloroform and the concentration of DPC in MeOH varies from 0 to 5 mg/mL to obtain the optimized thickness. During the deposition process of the active layer, the contact time between DPC and solution of active layer was critical and was reduced by dynamic spin coating method to prevent potential erosion of DPC underlayer (that is, 18 μ L of the active layer solution rapidly dripped onto the substrate while the CIL coated substrate was rotating at high speed). The detailed results were summarized in Table R2 and Supplementary Table 5. In inverted structure, device without DPC CIL shows an efficiency of 0.5% with large decrease in V_{OC} and FF, implying inefficient charge transport inside the device. The optimized device with DPC layer formed from 1 mg/ml solution exhibits the highest efficiency of 13.7% for single-junction and 21.7% for corresponding perovskite/OSC tandem solar cell.

We also added “the DPC device with optimized concentration of 1 mg mL⁻¹ delivered a V_{OC} of 0.85 V, J_{SC} of 22.86 mA cm⁻² with the optimal PCE of 13.7%.” in the revised manuscript.

Table R2. The efficiency of D18:L8-BO based device with different concentration of DPC as CIL.

Con. of DPC (mg mL ⁻¹)	J_{sc} (mA cm ⁻²)	V_{oc} (V)	FF (%)	PCE (%)	$J_{cal, EQE}$ (mA cm ⁻²)
0	20.26	0.084	27.94	0.5	20.14
0.5	21.90	0.854	62.36	11.7	20.67
1	22.85	0.851	70.4	13.7	21.89
2	21.90	0.853	58.71	11.0	21.15
3	21.32	0.831	51.90	9.2	20.87
5	21.07	0.819	43.71	7.5	20.58

3. How about the conductivity of DPC? Is there a self-doping effect in the DPC?

Response: We have added following discussion in revised paper according to the reviewer's concerns.

The self-doping effect of CIMs were investigated by the electron spin resonance (ESR) spectroscopy that can be seen from Fig. R3a and Supplementary Fig. 14 in revised paper. We firstly investigated the sole component of BCP, SPC and DPC. Very weak signal around g value of 2 from unpaired electron were detected in BCP, SPC and DPC. To further resealing the self-doping effect happed between acceptor and CIM, we also tested the ESR spectra of L8-BO:CIMs. The sample was prepared by dissolving L8-BO and CIM with molar ratio of 1:1 in chloroform followed by drying in a vacuum chamber. From the result, the L8-BO:DPC inhabits the highest resonance peak, suggesting the most efficient electron transport when compared with L8-BO:SPC and L8-BO:BCP.

We have measured the conductivity of DPC in a sandwiched device with structures of

ITO/CIL/Ag as shown in Fig. R3b (Fig. 1c in maintext). The conductivity values of BCP, SPC and DPC were obtained from the slope of I - V plot according to the previous report (*Energy Environ. Sci.* 2015, **8**, 1602-1608). As we mentioned in the maintext “The σ values of BCP, SPC and DPC were 0.0013, 0.0027 and 0.0036 mS cm^{-1} , respectively. The DPC exhibits the largest σ value indicating the highest vertical conductivity which is good for fast transporting the electron as soon as photogenerated carriers are separated.”

The electron mobility of BCP, SPC and DPC were quantified according to the space-charge-limited current (SCLC) method by measuring electron-only devices with a structure of ITO/ZnO/L8-BO/CIL or CIL free/Ag. As shown in Figure R3c (Supplementary Fig. 15 in the maintext), “pure L8-BO film presents a relatively low electron mobility of $1.4 \times 10^{-4} \text{ cm}^2 \text{ V}^{-1} \text{ s}^{-1}$. However, the device with DPC as CIL achieves an enhanced mobility of $6.9 \times 10^{-4} \text{ cm}^2 \text{ V}^{-1} \text{ s}^{-1}$, which is higher than the values for devices based on SPC ($4.8 \times 10^{-4} \text{ cm}^2 \text{ V}^{-1} \text{ s}^{-1}$) or BCP ($4.3 \times 10^{-4} \text{ cm}^2 \text{ V}^{-1} \text{ s}^{-1}$). The higher charge transport capability also accounts for the improvement of photovoltaic performance for DPC based solar devices.”

Figure R3. (a) ESP spectra of the solid samples of BCP, SPC, DPC, L8-BO:BCP, L8-BO:SPC and L8-BO:DPC; (b) The conductivity of various CILs measured by I - V curves with a sandwich structure of ITO/BCP/Ag, ITO/ SPC/Ag and ITO/DPC/Ag. (c) The electron mobility of L8-BO covered with CIL of BCP, SPC and DPC or CIL free.

4. Please also provide illumination data of the tandem devices.

Response: We have added the illumination stability of BCP and DPC CIL based TSC devices, which were tested under a LED light with intensity of 1 sun in glovebox as

shown in Figure R4a. After been photoaged under continuous light for 500 h, the BCP based device dropped to 77.8% of its initial efficiency while the DPC based device can maintained 94.7% of its initial PCE, indicating better illumination stability of DPC TSC. Due to limited testing time, we further tested the device efficiency of DPC-based TSC before and after been placed under ambient light in glove box for 3500 h. The result indicates that the DPC based TSC maintained a good stability in N₂ under ambient light, with an initial PCE of 20.3% and a final value of 18.2%.

Figure R4. (a) Illumination stability of TSC device based on BCP and DPC CIL that tested under a 1 sun LED light for 500 h in glovebox, (b) The *J-V* curve of DPC TSC that before and after been kept in the glove box under ambient light for 3500 h.

5. DPC has strong absorption in the visible range. How will it influence the device performance?

Response: We appreciate the reviewer for this comment. The absorption spectra of SPC and DPC single layer are shown in Fig. R5a (Fig. 1a in maintext), with the relevant descriptions also presented in the corresponding positions of the maintext. The absorption peak of SPC is located at 543 nm and the peak of DPC red shifted to 558 nm because of the added carbonyl substitution.

For the single junction OSC device with a conventional structure as shown in Fig. R5b (Fig. 2a in the maintext). Within such device, light will first pass through the anode and the active layer, after being partially absorbed, the remaining below bandgap light will reach DPC layer and cathode, therefore except for negligible influence on electrode reflection light, the absorption of DPC has imperceptible effect on the active layer and corresponding device performance.

While in perovskite/OSC tandem solar cells, with device structure presented in Fig. R5c (Fig. 5a in maintext), the front DPC layer does have absorption overlap with the organic layer. However, as is shown in Fig. R5d (Fig. 5d in maintext), light with energy above the bandgap of CsPbI₂Br (~1.92 eV, i.e., light above 650 nm) was adsorbed by the perovskite layer, leaving light shorter than 650 nm being utilized by the organic D18:L8-BO layer, which DPC barely absorb according to transmittance spectra in Fig. R5e. Therefore, the light utilization in tandem device was precisely controlled, there is no light absorption competition between DPC and organic absorber. Therefore, no matter in the conventional single junction organic device or perovskite/OSC tandem device, DPC has negligible effect on the light absorption of photovoltaic layer.

We have added the transmittance data (Supplementary Fig. 25 in revised maintext) and corresponding description is shown as below.

“For perovskite/OSC TSC, the ICL beneath active layer of rear cell should have a high optical transmittance to ensure that more transmitted light can be utilized by rear-cell. Therefore, we measured the transmittance of BCP and DPC film, with data depicted in Supplementary Fig. 25. Although BCP film has very high transmittance between 300 to 800 nm, the DPC film obtained with optimized concentration (1 mg/mL) shows a very high transmittance of more than 97% in the long-wavelength region from 400 nm to 1000 nm, which is good enough for the solar cell utilization. Consequently, the high transmittance of BCP and DPC can ensure adequate light-harvesting for photoactive layer of tandem solar cell.”

Figure R5. (a) absorption spectrum of SPC and DPC, (b) device structure of single

junction OSC, (c) crossing section-SEM image of perovskite/OSC tandem solar cell, (d) EQE spectra for perovskite/organic tandem solar cell, (e) transmittance of the BCP and DPC film.

6. Please comment any specific reason of using the carbonyl unit for designing the DCP.

Response: There are two mainly reasons when we use carbonyl units for DPC molecular design.

1) According to previous reports, the metal-carbon triple bonds and carbon-carbon triple based $d\pi-p\pi$ conjugated system can work well as electron transport materials for both organic solar cells and perovskite solar cells (*Nat. Commun.* **2020**, *11*, 4651). Systematic studies have found that when π -conjugation skeleton was introduced to a conjugated side chain can strengthen intermolecular charge transfer and make the electron transport orderly in device (*Adv. Mater.* **2021**, *33*, 2101279). Furthermore, the dipole structure of such materials can reduce the work function of metal electrode and finally improved the device efficiency and stability (*J. Am. Chem. Soc.* **2021**, *143*, 7759–7768). Therefore, introducing carbonyl into the conjugate skeleton is a very feasible strategy to design highly efficient electron transport materials.

2) Electron transport materials containing amines or N atoms have been proved that it's very good for intramolecular charge transport and promoting device efficiency (*Adv. Func. Mater.* **2022**, *32*, 2111706, *Acs Energy Lett.* **2022**, *7*, 4052-4060). However, this type of materials act as nucleophile and can react with C=C double bond within non-fullerene acceptors, which is harmful to the long-term stability of corresponding device.

Similar situation happens to BCP (*J. Mater. Chem. A*, **2021**, *9*, 23269–23275) which is very commonly used interfacial material for highly efficient perovskite solar cells. Therefore, we chose the Phen core of BCP material and connected it with carbonyl unit. According to the large steric hindrance and strong electron absorption ability of carbonyl, the as created material was designed to be highly efficient with reduced reactivity of N atoms and non-fullerene acceptor materials.

Reviewer #2 (Remarks to the Author):

For the performance of highly efficient and stable organic solar cells cathode interfacial layers play an important role. They transfer the negative charges and block the positive charges at the cathode. In the present manuscript, the authors report a on a variety of cathode interface materials that are available for both efficient single junction organic solar cell and tandem solar cell. According to the rational molecular design strategy of carbonyl substituents on the Phen, the photovoltaic performance of resulted DPC and SPC improved significantly, such as higher electrical conductivity, better matching energy levels or suitable work function modification of metal cathode. As a result, the single junction organic solar cell obtained an OPV efficiency of 18.2% which is higher than that of BCP control device.

The authors found that DPC with two carbonyl substitutions can suppress the negative reaction which commonly happens between the cathode interfacial layer and non-fullerene acceptor (NFA) material, leading to enhanced device stability. In particular, the DPC based device maintained 80% of its initial PCE at elevated temperatures (85°C) for 96 h or at room temperature after 2170 h in inert N₂ atmosphere. As the reported DPC material exhibits overall better photovoltaic performance and stability, DPC offers a wide range of potential of applications in OPV community.

Just from these novel achievements, I do recommend the manuscript for publication in Nature Communications. However, there are several aspects that have to be changed/clarified/commented prior to final acceptance.

We thank reviewer for the insightful and valuable comments and suggestions.

1. There are too many experimental findings listed in the introduction part. Authors should just name the 3 key findings.

Response: We thank the reviewer for the suggestion. We reduced the description of research progress in the introduction, but kept the most important reports of amine

group containing CIMs which are highly related to our research work.

“Many efforts have been carried out to overcome this interfacial instability by developing more stable CIMs. To minimize the reactivity of the amine groups in CIMs, Xiong *et al.*¹⁰ proposed a method to protonate the amine group in PEIE to discourage the interfacial reaction and achieved a flexible NFA OSC with a power conversion efficiency (PCE) of 12.5%. Qin *et al.*¹⁶ found chelation of PEI with Zn²⁺ (PEI-Zn) can be realized by addition of zinc acetate dihydrate into the PEI solution. This allows strong chelation between Zn²⁺ and amino groups, thus inhibiting the reaction between PEI and the IT-4F. In this way, chelation of metallic elements was shown to be another way to design and synthesize high performance CIMs with good device efficiency and stability.”

2. Materials characterization section: the achieved data should be summarized in one common table (which also allows better comparison of the materials' properties).

Response: We thank the reviewer for the helpful comment. We have summarized the photophysical properties of BCP, SPC and DPC in one table and added it in the revised manuscript.

Table R3. The photophysical properties of BCP, SPC and DPC.

CILs	Film absorption (nm)		Conductivity mS cm ⁻¹	LUMO (eV)	HOMO (eV)	WF of Ag with CIL (eV)
	λ_{\max}	λ_{edge}				
BCP	/	400	0.0013	-3.12	-6.22	3.95
SPC	543	611	0.0027	-4.45	-6.48	3.80
DPC	558	685	0.0036	-4.69	-6.50	3.55

3. Fig. 2d: kinetic energy scale in UPS does not give any information if now experimental details are declared. Actually, all relevant data are shown in Fig. S11 (with

all required information on determination of the respective values (slopes and crossing with energy axis)

Response: We have added the auxiliary lines necessary to obtain the data in Fig. 1d, and the specific values are presented in the maintext. Because the high conductivity of Ag electrode, the Fermi level is set to zero in binding energy axis, therefore the work functions of Ag and Ag/CIL can be determined simply by subtracting the energy difference between Fermi energy and cut-off energy with 21.2 eV (He I). We take the subtracted value and re-drew the UPS spectra as shown in Fig. R6 and Fig. 1d in revised paper.

Figure R6. UPS spectra of Ag before and after being coated with BCP, SPC and DPC CIL.

4. HOMO and LUMO orbitals are important for material analysis. The authors may add the simulated frontier molecular orbitals to the supplementary information. The energy level calculation method should be described in the experimental section of the manuscript.

Response: We have added following discussion in revised paper according to the reviewer's suggestion.

“The simulated distribution of HOMO and LUMO orbitals was provided in Fig. R7 (Supplementary Fig. 12 in the revised paper). There is strong π -delocalization between the Phen core and carbolong frameworks in SPC and DPC.”

Figure R7. DFT calculations of (a) LUMO energy level and (b) HOMO energy level of BCP, SPC and DPC (from left to right in each panel)

5. Degradation studies (Figure 3). It is not fully clear how relevant the XPS study is. Is it required to show the materials' degradation in the main text. Major conclusion is that BCP shows significantly different composition (according to N1s binding energies), but the conclusions is more speculative.

Response: To better explain the relevance of XPS measurement, we would like to give more detailed explanation based on the content of the revised manuscript.

1) To deeply understand the different chemical reactivity between BCP, SPC and DPC with L8-BO, we have carried out electrostatic potential (ESP) to study the nucleophilic ability of those three CIMs, which is shown in Fig. 2d (in the maintext). When compared with BCP, “the extremely small ESP value around the N atom of DPC indicates that the DPC has the weakest nucleophilic ability and thus greatly reduces the chemical reactivity between the acceptor at the interface.”

2) To reveal the materials' degradation process, we measured ^1H NMR spectra of pure L8-BO, BCP and DPC solution as well as the mix solution of L8-BO:BCP and L8-BO:DPC under $80\text{ }^\circ\text{C}$ (Fig. 2e and 2f in maintext). We found that “after the L8-BO:BCP mixed solution was heated for 2 h, the characteristic H signal in the C=C peak split to two peaks indicating that the chemical structure of L8-BO began to destroy during the heating process.” However, it was confirmed that DPC could greatly suppress such

chemical reaction when the Phen core connects with two carbonyl units in the structure (Fig. 2f in maintext).

3) The final chemical product of BCP and L8-BO was characterized as shown in Supplementary Fig. 15 and Fig. 16 in maintext. The degradation of L8-BO when it was attracted by BCP was described in maintext as “it was highly possible that the nitrogen atoms on the BCP react rapidly with the C=C double bond of L8-BO and then destroy its structure.”

Therefore, we have clearly explored the chemical reaction process that happened between BCP and L8-BO, which is the nucleophilic attack of the N atom in BCP to NFA materials. In the illumination stability part, we conducted XPS measurement by checking the chemical status of nitrogen atom in BCP or DPC to study the illumination stability of devices with different CILs. Considering the reviewer’s comments and making this experiment more clearly, we added samples of L8-BO/BCP and L8-BO/DPC film without any photoaging for comparison, the results are depicted in Fig. R8, and Fig. 3b, 3c in the revised manuscript. The related discussion has been expanded/replaced as following:

“To deeply insight into the completely different effect of BCP and DPC on illumination stability of device, we carried out X-ray photoelectron spectroscopy (XPS) measurements to trace the signal of N 1s orbital. In this measurement, we firstly tested the N 1s signal of pure BCP or DPC CIL film without any photoaging as shown in Fig. R8a and 8b (supplementary Fig. 21 in the revised paper). The characteristic N 1s peak of fresh BCP is located at 398.5 eV while DPC shifted to 399.5 eV due to the two carbonyl units with strong electron withdrawing property on the structure. For the fresh BCP or DPC film that spin-coated upon L8-BO film as shown in Fig. R8c and d (Fig. 3b and c in revised manuscript), the XPS peak of N 1s are situated at 398.5 and 399.5 eV, respectively, which are same locations with the pure BCP and DPC film. After been illuminated for 2000 h, the N 1s peak of DPC that covering upon L8-BO layer also show a stronger peak at 399.5 eV which is same location as pure DPC film. However, the peak of L8-BO/BCP film moved to 398.9 indicating that the chemical structural of BCP changed during the photon-oxidation process.”

Figure R8. The XPS spectra characterization of the signal from the N 1s orbital of (a) fresh BCP, (b) fresh DPC film, (c) L8-BO/BCP, (d) L8-BO/DPC with or without being illuminated for 2000 h.

6. Page 19, line 340: is the variation in the 3 digit really trustworthy?

Response: From the photovoltaic parameters that summarized in Table 1 of maintext, the devices based on BCP, SPC and DPC CIL can achieve relatively high device efficiency (more than 17%), which means the carrier separation efficiency inside those three devices are good, suggesting very similar level of bimolecular recombination. To answer this question, we further measured transient photocurrent (TPC) and added the following discussion in revised manuscript.

In order to further explore the exciton separation in devices with BCP, SPC or DPC CIM, we conducted transient photocurrent (TPC) measurement as shown in Fig. R9 (Supplementary Fig. 23 in revised maintext). The fitted decay time of BCP, SPC and DPC based devices are 0.55, 0.23 and 0.22 μ s, respectively. The most reduced extraction time confirmed the superior charge extraction ability of DPC based device.

Figure R9. TPC measurement of device based on different CILs.

7. Results reported on page 22 just report on the device parameters – authors should focus on the most prominent ones required for their discussion/conclusions.

Response: We thank the reviewer for this insightful comment. As we have previously discussed about the inherent robustness and restricted chemical reaction of DPC layer, DPC was sandwiched between Ag and organic BHJ layer in tandem device structure, the high thermal stability of DPC enables thermal treatment of organic above layer to achieve better crystalline and improved photovoltaic performance.

We have revised this part and made it more consistent and logical with previous discussions.

The photovoltaic parameters of individual single-junction sub-cells are summarized in Supplementary Table 5. The CsPbI₂Br device, which features an optical bandgap of 1.92 eV, was used to absorb photons with higher energy in tandem devices. Firstly, the BCP and DPC based devices were prepared with an inverted structure of ITO/BCP (DPC)/D18:L8-BO/MoO₃/Ag, that were exactly same as rear cell in the tandem device to fully simulate the performance. BCP device with an inverted structure only obtained a PCE of 3.5% which is much lower than BCP-based OSC with a conventional structure. As a result, the PCE of corresponding BCP-TSC only achieved 12.4%. On the contrary,

give the credit to strong electron-withdrawing properties and large steric hindrance of the carbonyl units, the DPC device with optimized concentration of 1 mg/mL delivered a V_{oc} of 0.85 V, J_{sc} of 22.86 mA cm⁻² with the optimal PCE of 13.7%. More importantly, the resulted DPC-TSC achieved an impressive PCE of 21.7% with a high V_{oc} of 2.07 V, a J_{sc} of 12.95 mA cm⁻² and an astonishing FF of 80.8%. Fig. 5c presents the EQE spectra of DPC based TSC measured sequentially applying the bias light with 550 nm short pass filter and 800 nm long pass filter. The evenly split spectra of front cell and rear cell generated well matched current owing to the precisely controlled thickness of absorber. Besides, a high average PCE of 20.8% that obtained from 30 cells in Fig. 5e indicates the high repeatability of DPC based TSC. We further tested the thermal stability of tandem device, and the results are shown in Fig. 5f. Due to the suppressed chemical activity and inherent stability, the DPC based device maintained a good stability during storage in N₂ glovebox under 85 °C continues heating for 90 h, 80.0% of its initial PCE remained, while the BCP based device loses its $J-V$ characteristic after the same time. The outstanding device performance and enhanced stability of DPC as ICL for tandem solar cells indicating the potential wide application of carbonyl in the photovoltaic field.

8. Can the authors give any idea why DPC and SPC are advantageous to suppress interfacial reactions when used as ICL?

Response: We believe the suppressed interfacial reaction within DPC and SPC device originated from the inherent inertness of our novel designed molecule structure. In comparison with widely used BCP, the addition of carbonyl unit for SPC and DPC, which presents strong electron absorption and large steric hindrance, can effectively protect the contact between nitrogen atoms in Phenanthroline (Phen) core and nucleophilic sites in acceptor materials. It can be further confirmed by ESP spectra of BCP, SPC, and DPC in Figure 2d (maintext). Among these three molecules, the electron-rich region is mainly located around N atoms of Phen, with the fact that the lowest ESP value of DPC indicates that the DPC has the weakest nucleophilic ability and thus greatly reduces the chemical reactivity with C=C bonds in NFA.

9. Authors should use the identical color code for BCP, SPC and DPC in all figures.

Response: We have made modifications in main text and supporting information and unified the color of curves for BCP, SCP and DPC with color of orange, green and blue, respectively.

10. Some abbreviations must be explained at their first appearance (NB: no abbreviations in the abstract).

Response: We have carefully checked the abbreviations in the revised manuscript.

11. The manuscript may require some rewriting/rephrasing and inspection from a native English speaking person. The use of the article “the” sometimes seems arbitrary. The word “meanwhile” is not used in the proper sense. I also found few incomplete sentences and numerous typos). See e.g. Fig. S20 (“Bing Energy”)

Response: According to the reviewer’s suggestion, we have carefully checked the typos and grammar issues in the revised manuscript.

12. Line 385: Supplementary “Table 3.” should be corrected to “Table 4”.

Response: We have carefully checked and corrected these mistakes in the revised manuscript.

Reviewer #3 (Remarks to the Author):

In this work, the authors developed two newly synthesized cathode interfacial materials, DPC and SPC, which consist of phenanthroline core and carbonyl unit. The authors investigated their improved photovoltaic performance and utilized them in organic solar cells as well as perovskite-organic tandem solar cells. They used various methods to evaluate and compare the effect of carbonyl CIMs on device efficiency and stability and confirmed that the resulting DPC can minimize the chemical reaction which indeed plays an important role in the long-term stability of corresponding photovoltaic devices. Overall, this work provides lots of valuable information for molecular design.

Publication of this work is recommended in Nature Communications after minor revisions.

Thank the reviewer for this positive comments.

1) In Fig.4b, the authors investigate the charge collection efficiency (η_c) of device based on various cathode interfacial layers. However, the obtained values of DPC, SPC, and BCP are too close to distinctly compare the charge transfer ability at this layer. The authors should provide more evidence on this point.

Response: we thank the reviewer for this comment. To give more information about charge transport and collection of devices based on various CIMs, we further tested conducted transient photocurrent (TPC) measurement which allows to study time-dependent extraction of charge, the result is shown in Fig. R8 and Supplementary Fig. 23 in revised manuscript. We added the following discussion in the revised manuscript.

“In order to further explore the exciton separation in devices with BCP, SPC or DPC CIM, we conducted transient photocurrent (TPC) measurement as shown in Supplementary Fig. 23. The fitted decay time of BCP, SPC and DPC based devices are 0.55, 0.23 and 0.22 μ s, respectively. The most reduced extraction time confirmed the superior charge extraction ability of DPC based device”

2) What is the essential reason for the strong electron-withdrawing properties of carbolong frameworks?

Response: The charged triphenylphosphonium substituents and the electron-deficient nature of transition metals are the key for the strong electron-withdrawing properties of carbolong frameworks. To get more insight, we performed DFT calculations as shown in Fig. R10. The structure of **DPC-H**, which has hydrogen atoms replacing the charged triphenylphosphonium substituents in **DPC**, shows a large ESP value around the N atoms. It suggests that the charged triphenylphosphonium substituents exhibit strong electron-withdrawing ability. On the other hand, the transition metal osmium features electron-deficient characteristics.

Figure R10. The ESP of **DPC** and **DPC-H**.

3) Organometallic compounds generally suffer from poor stability. However, why the carbonyl complexes are quite stable?

Response: The carbonyl complexes are in aromatic systems (*Nat. Chem.* **2013**, *5*, 698–703), the intrinsic aromaticity together with the rigid pincer frameworks can stabilize the cyclic osmium species (*iScience* **2019**, *19*, 1214–1224), which makes the carbonyl complexed quite stable.

4) The addition reaction of alkynes and metal carbynes is interesting. It is better to add some discussion about this reaction.

Response: Thank the reviewer for the suggestions. We have added following discussions on this addition reaction to the revised manuscript.

“It is is the electrophilic addition of carbyne by the alkyne under the synergistic effect of H^+ .”

5) Acronyms for nouns should appear where they are first mentioned in the manuscript,

including but not limited to BCP, CIM and TPC.

Response: We have carefully checked the acronyms and corrected in the revised manuscript.

6) The energy levels of commonly used materials in OPV field are summarized in supplementary figure 12, references are needed here.

Response: We thank the reviewer for this comment. We have summarized references in Supplementary Table 2 in the revised manuscript.

7) Page 26 line 461, reference 42 is wrong.

Response: We thank the reviewer for pointing out this mistake, which has been corrected in the revised manuscript.

8) In Scheme 1, what is the abbreviation of [Os]? please specify.

Response: The abbreviation of [Os] is $\text{OsCl}(\text{PPh}_3)_2$. We have added the explanatory note to the Scheme 1.

We are very grateful for all the suggestions from the reviewers, which have improved our manuscript significantly.

REVIEWERS' COMMENTS

Reviewer #1 (Remarks to the Author):

The authors have addressed the questions and concerns. It could be accepted at this moment.

Reviewer #2 (Remarks to the Author):

The authors have carefully responded to the comments of the 3 reviewers and addressed all major concerns in their revised manuscript.

I therefore suggest acceptance of the revised manuscript in its present form.

Reviewer #3 (Remarks to the Author):

The revised manuscript has fully addressed all comments from reviewers, therefore, I recommend the publish of this work without further revisions.